# FlashHead: Efficient Drop-In Replacement for the Classification Head in Language Model Inference

## Abstract

Language models are increasingly adopting smaller architectures optimized for consumer devices. In this setting, inference efficiency is the primary constraint. Meanwhile, vocabulary sizes continue to grow rapidly, making the classification head a critical bottleneck that accounts for up to 60% of model parameters, and 50% of inference compute. We introduce FlashHead, the first efficient drop-in replacement for the dense classification head that is training-free and hardware-friendly. FlashHead builds on principles from information retrieval, where the output head is viewing the output head as a retrieval problem rather than a dense computation. FlashHead introduces four key innovations: (1) equal-sized clustering of embeddings, (2) multi-probe retrieval to model heads, (3) a novel inference-time sampling mechanism, and (4) selective quantization, enabling effective low-bit computation in the head. Experiments on Llama-3.2, Gemma-3, and Qwen-3 show that FlashHead delivers model-level inference speedups of up to 1.75x while maintaining output accuracy compared to the original head. By overcoming the classification head bottleneck, FlashHead establishes a new benchmark for efficient inference and removes a key barrier to developing smaller, capable models for consumer hardware.

## 1 Introduction

As small language models (SLMs) become central to wide adoption of machine learning systems across society, their deployment on consumer-grade hardware is becoming increasingly important Belcak et al. (2025); Shakhadri et al. (2024); Lu et al. (2025); Zhang et al. (2025). In state-of-the-art SLM architectures such as Llama-3.2 Research (2024), Gemma 3 Team (2025), and Qwen 3 Yang et al. (2024), the classification head alone accounts for between 20% and 60% of total model parameters and dominates inference cost. As the general trend is toward increasing vocabulary sizes Tao et al. (2024), this problem is likely to be a central challenge for efficient inference in SLMs.

Prior to the recent emphasis on SLMs, research on efficient deployment of large language models concentrated almost entirely on the transformer body. Quantization techniques Frantar et al. (2022); Lee et al. (2024); Lin et al. (2024) were developed to compress weights and activations while maintaining model accuracy, yet these methods explicitly avoid quantizing the classification head. This limitation is also reflected in inference frameworks like vLLM Kwon et al. (2023), which as of the latest release leave the output head in higher precision to avoid accuracy degradation.

Prior work targeting the model head largely predates the current emphasis on SLMs, and relatively few approaches have been designed with inference efficiency as the primary goal. Proposed methods span hierarchical softmax Morin & Bengio (2005) and vocabulary-trimming strategies Ushio et al. (2023), to adaptive output layers Zhao et al. (2021). However, these typically require additional fine-tuning Mnih & Hinton (2008); Morin & Bengio (2005); Joulin et al. (2017); Chen et al. (2015; 2025); Zhao et al. (2021), task-specific calibration Ushio et al. (2023), or fail to faithfully capture the full output probability distribution Shim et al. (2017); Zhang et al. (2018).

At their core, these methods view the classification head as a two-stage retrieval mechanism. One family of approaches replaces the dense head with Approximate Nearest Neighbor (ANN) algorithms Zhang et al. (2018); Chen et al. (2025); Zhao et al. (2021). Variants of the inverted file (IVF)

index are commonly used, and several methods Chen et al. (2025); Zhao et al. (2021); Chen et al. (2018) explicitly exploit spherical K-Means Dhillon & Modha (2001) to align with the natural semantic structure of language models Petukhova et al. (2025). Despite their promise, these methods inherit the same drawbacks noted above.

Motivated by these limitations, we propose FlashHead: the first efficient drop-in replacement for the dense classification head that is both training-free and hardware-friendly. FlashHead introduces four key innovations:

1. **Equal-sized clustering**: We employ a variant of spherical K-means that enforces strictly equal cluster sizes. This design allows centroids to be packed into compact tensors, enabling fast and predictable memory access. To our knowledge, FlashHead is the first LLM head that clusters token embeddings into strictly equal-sized spherical clusters.

2. **Efficient multi-probing**: Whereas prior work typically restricts retrieval to the single most likely cluster, our compact clustering enables aggressive use of multi-probing Lv et al. (2007). We depart significantly from existing applications of information retrieval in language models by scaling from hundreds of clusters to tens of thousands. We achieve this by employing hundreds to thousands of probes to simultaneously score several clusters. This expanded probing strategy, designed specifically for inference in language models, is highly accurate while remaining efficient.

3. **Probabilistic probe sampling**: Unlike hierarchical softmax, which factorizes and learns the probability distribution end-to-end, our method naturally bridges retrieval and probabilistic decoding. We introduce a multinomial sampling step at the probe-selection stage, allowing top-token retrieval and sampling to be seamlessly unified.

4. **Selective quantization**: FlashHead admits coarse quantization in the first stage of its retrieval process. By distributing probability mass across such a large number of clusters and probes, it becomes naturally robust to quantization. This makes FlashHead the first method to enable safe and effective quantization of the output head. Flashhead in higher precision outperforms quantization of the dense head, and in combination with quantization delivers even greater performance.

The four components are not independent optimizations but synergistic mechanisms, whose combination is crucial for the overall improvements, as demonstrated in our experiments. Overall, Flash-Head turns the output head from a dominant bottleneck into a negligible component of inference. Crucially, the gains are not just at the head level: across all benchmarked models we see consistent *model-level* speedups. For instance, FlashHead accelerates Llama-3.2-1B by up to 1.75×, Gemma-3-270M by 1.35×, and Qwen-3-1.7B by 1.31×, all without retraining and marginal accuracy loss. Even at the 8B scale, FlashHead delivers a 1.13× speedup where baseline inference already saturates consumer GPUs. In effect, what previously consumed a large portion of compute is compressed into an insignificant fraction, enabling state-of-the-art SLMs to run faster and leaner on commodity hardware.

Our main contributions can be summarized as follows:

1. We introduce FlashHead, the first training-free and hardware-friendly drop-in replacement for the dense classification head, which substantially improves inference efficiency while preserving accuracy.

2. We provide a comprehensive evaluation of FlashHead against existing alternatives, covering reasoning, multilingual, and instruction-following benchmarks across multiple model families, as well as detailed on-device latency measurements.

3. We release a public implementation of FlashHead integrated into several state-of-the-art models, including Llama 3.2, Gemma 3, and Qwen 3 (available at [REDACTED URL DUE TO BLIND REVIEW]), to facilitate adoption and further research.

## 2 RELATED WORK

Research on efficient alternatives to the dense classification head falls into two categories: trainable replacements and training-free replacements.

## 2.1 TRAINABLE REPLACEMENTS

Several works replace the dense head with structures that must be retrained or fine-tuned alongside the model. Hierarchical softmax organizes the vocabulary into a tree to reduce complexity, but suffers from tree construction bias and limited flexibility Morin & Bengio (2005). Adaptive softmax groups tokens by frequency and learns cluster heads jointly with the model Joulin et al. (2017), while differentiated softmax allocates higher-rank embeddings to frequent tokens and lower-rank ones to rare tokens Chen et al. (2015).

Prior work has also considered spherical K-means clustering of context vectors as a retrieval mechanism for inference Chen et al. (2018). These methods partition queries rather than keys and thus depend on training-time statistics to define candidate sets. ANN-based training methods such as ANN-Softmax Chen et al. (2025) or MIDX Sampler Zhao et al. (2021) adopt inverted-file or product-quantization indexes also based on K-means to accelerate training.

While several of these approaches rely on K-means to partition the vocabulary, they cluster queries (context vectors) and depend on training-time statistics or retraining to define candidate sets. Flash-Head clusters the keys (token embeddings) once, offline, and introduces several innovations that enable a novel multi-probe retrieval, preserving the full probability distribution without any retraining or data.

## 2.2 TRAINING-FREE REPLACEMENTS

Training-free replacements avoid retraining and therefore represent the most practical deployment alternatives. Vocabulary trimming Ushio et al. (2023) discards infrequent tokens using a calibration set, offering speedups but fundamentally restricting the vocabulary and reducing robustness to rare or out-of-distribution prompts. SVD-Softmax Shim et al. (2017) pioneered low-rank decompositions for accelerating large-vocabulary softmax layers. However, because sampling occurs only after the coarse approximation step, probabilities are reliable primarily for high-likelihood tokens. Fast Graph Decoder Zhang et al. (2018) introduced approximate nearest neighbor search as a training-free replacement for classification heads. Its formulation, however, outputs only a top-$k$ candidate set rather than modeling the full probability distribution, limiting its applicability. With the exception of vocabulary trimming, these works were developed before the transformer era, when recurrent and embedding-based models dominated.

FlashHead belongs to the training-free category, while addressing existing limitations. In our experiments we therefore compare primarily to existing training-free replacements, which represent the most practical deployment alternatives.

## 3 METHODOLOGY

In LLMs, the dense classification head transforms a $d$-dimensional hidden state vector $\mathbf{h} \in \mathbb{R}^d$ (where $d$ is the embedding size) to a vocabulary-sized logit vector. The hidden state vector encodes the contextual state produced by the model body after processing an input sequence. Concretely, the classification head applies a single matrix multiplication: $\mathbf{z} = \mathbf{E} \times \mathbf{h}$, where $\mathbf{E} \in \mathbb{R}^{v \times d}$ is the output-embedding matrix (with $v$ being the total number of tokens in the vocabulary) to produce the logit vector. Each row $\mathbf{e}_i$ is the $d$-dimensional embedding of token $i$.

For greedy retrieval the next token is selected as $t = argmax(\mathbf{z})$, i.e. the index of the largest logit in $z$.

For probabilistic token sampling, the logits are scaled by a temperature $\tau > 0$ and converted to a distribution

$$\mathbf{y} = \text{softmax}\left(\tfrac{\mathbf{z}}{\tau}\right), \quad t \sim \text{Categorical}(\mathbf{y}).$$

## 3.1 CLUSTERING THE EMBEDDING MATRIX

In FlashHead, a one-time offline clustering step is applied to partition the $v$ embedding vectors of $\mathbf{E} \in \mathbb{R}^{v \times d}$ into $c \ll v$ disjoint clusters $\{\mathcal{C}_1, \ldots, \mathcal{C}_c\}$. Specifically, *spherical $k$-means* Dhillon & Modha (2001), a variant of $k$-means that measures similarity with the cosine metric, is applied.

Cosine distance is a natural choice because empirical studies show that the semantic information in token embeddings is largely encoded in their *direction*, with vector length playing only a minor role Mikolov et al. (2013).

**Objective.** Given a number of clusters $c$, spherical *k*-means minimizes the negative cosine similarity between points and their assigned centroids Dhillon & Modha (2001):

$$\min_{\{\mathcal{C}_k\}} \sum_{k=1}^{c} \sum_{i \in \mathcal{C}_k} \left(1 - \mathbf{e}_i^\top \mathbf{c}_k\right), \qquad \text{with} \quad \mathbf{c}_k = \frac{\sum_{i \in \mathcal{C}_k} \mathbf{e}_i}{\left\|\sum_{i \in \mathcal{C}_k} \mathbf{e}_i\right\|_2}.$$

Because every centroid is re-normalised after each batch update, the algorithm alternates between (i) assigning each token to the closest centroid on the sphere, and (ii) recomputing centroid directions as the mean of the cluster members.

We implement a modified assignment step that enforces an equal cluster size of $v/c$ tokens per cluster (thus $c$ is always a divisor of $v$). If a cluster exceeds its capacity, its lowest-similarity members are reassigned greedily to clusters that still have available slots, prioritizing the most similar eligible centroid. This ensures that all clusters have identical capacity while preserving semantic grouping. Enforcing *exactly* equal sized clusters is critical for efficiency as discussed below.

**Initialization and convergence.** Empirically we converge within a budget of 1000 iterations for all models evaluated. Clustering is a one-time cost and goes relatively fast. For example clustering the 128 256-token vocabulary of Llama-3.2 into $c = 8016$ clusters takes 4 hours on a single A40 GPU.

**Centroid matrix.** After clustering we store the normalized centroid matrix $\mathbf{C} \in \mathbb{R}^{c \times d}$. In addition, we require the mapping between clusters and token indices stored in a matrix $\mathbf{C2T} \in \mathbb{R}^{c \times b}$, where $b$ is the size of the clusters. When clusters are not equalized (for ablation studies), see 4.5, it represents the largest cluster (rows corresponding to smaller clusters are padded).

### 3.2 TOKEN RETRIEVAL FROM HIDDEN VECTOR

At inference time, FlashHead transforms the hidden state $\mathbf{h} \in \mathbb{R}^d$ into the next token via a dynamic two-step retrieval process inspired by the inverted file (IVF) index (see Algorithm 1). We support both decoding scenarios mentioned in Section 3; their key differences in FlashHead are illustrated in Figure 1.

**Deterministic / greedy decoding.** For tasks such as classification, after the matrix multiplication $\mathbf{C} \times \mathbf{h}$, the top $p$ centroids are selected given the produced centroid logits. FlashHead gathers tokens in these clusters into a reduced vocabulary embedding matrix $\tilde{\mathbf{E}}$. Given this embedding, token logits are computed over this smaller set, $\mathbf{z} = \tilde{\mathbf{E}} \times \mathbf{h}$, and the next token is chosen as $t = argmax(\mathbf{z})$.

**Stochastic / sampling decoding.** For open-ended generation, after the matrix multiplication $\mathbf{C} \times \mathbf{h}$, $p$ centroids are sampled without replacement, based on the softmax operation applied on the centroid logits. As for the deterministic/greedy decoding, FlashHead gathers tokens in these sampled clusters into a reduced vocabulary embedding matrix $\tilde{\mathbf{E}}$. In the second step, after the matrix multiplication $\tilde{\mathbf{E}} \times \mathbf{h}$, a token is sampled based on the softmax operation applied on the token logits.

**Complexity.** The dense head requires $O(vd)$ multiplications, where $v$ is the vocabulary size. FlashHead reduces this to

$$O(cd + pbd),$$

where $c$ is the number of clusters, $b = v/c$ the cluster size, and $p \ll c$ the number of probes used at inference. In practice, $p \cdot b \ll v$, yielding significant savings in both compute and memory access. Notably, as $c$ increases, $b$ decreases, and our design choices allows a practical implementation where we can exploit this to scale to very large $c$ and $p$ while remaining efficient. We demonstrate this scalability and its impact on accuracy–efficiency trade-offs in Section 4.5.

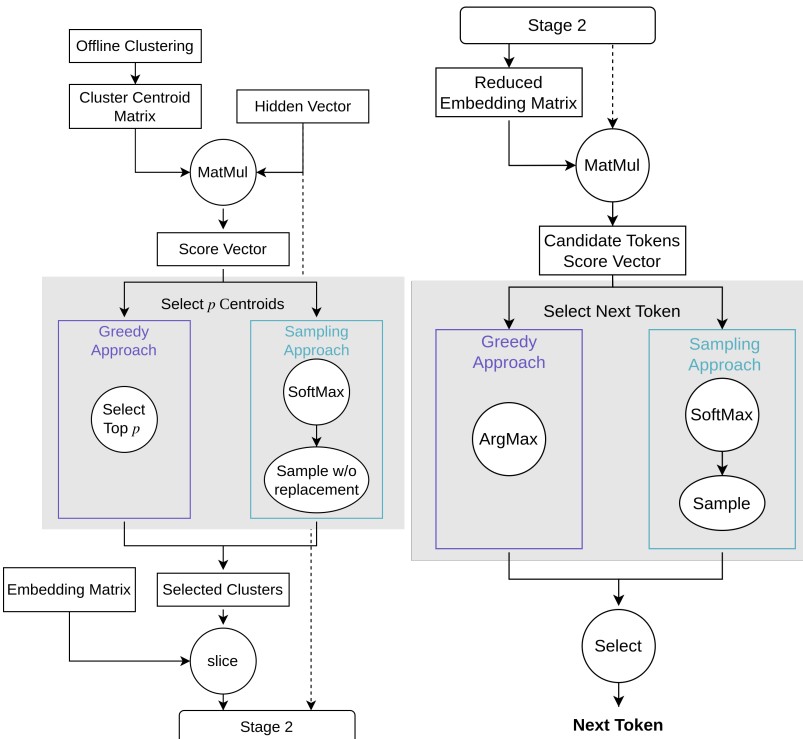

Figure 1: Several clusters of tokens are selected conditioned on the hidden vector produced by the model body. The reduced subset of token embeddings are multiplied with the hidden vector to make the final token selection.. Illustration of the proposed FlashHead algorithm, highlighting the difference between greedy approach of selecting the most likely next token, and the sampling approach allowing for probabilistic sampling of new tokens.

A critical source of efficiency comes from the cluster-to-token mapping **C2T**. Although not reflected in the multiplication count above, **C2T** governs how token embeddings are retrieved after probe selection. Enforcing equal cluster sizes allows **C2T** to be stored as a dense $[c \times b]$ tensor, so token indices can be computed with simple modular arithmetic. Without equal cluster sizes, **C2T** becomes ragged and requires per-cluster length tracking and masked gathers, which significantly increase memory access cost on accelerators. Section 4.5 shows that equal-sized clustering is essential to eliminate this overhead while preserving accuracy.

**Quantization note.** The first stage as described in Algorithm 1 (computing centroid logits $\mathbf{C} \times \mathbf{h}$) is straightforward to quantize because (i) it is a fully static matrix multiplication, and (ii) the coarse centroid probabilities are refined in the second stage using higher-precision token logits. As shown in Section 4.5, this selective quantization yields significant speedups while maintaining accuracy.

## 4 EXPERIMENTS

Evaluating language models (LMs) is nontrivial because open-ended generation intertwines correctness, fluency, and factuality Biderman et al. (2024); even small divergences in early tokens can cascade into valid but different continuations. We therefore report: (i) standard LM benchmark results on generation and reasoning tasks; (ii) *Top-k Containment* to isolate head-level fidelity; and (iii) on-device latency as time-per-output-token, both for the head alone ($\text{TPOT}^{\text{H}}$) and the full model (TPOT).

**Models, datasets, and harness.** We conduct extensive evaluations across several popular families of models: Llama-3.2, Llama-3.1, Qwen-3, Gemma-3. Unless stated, the base model is `Llama-3.2-1B-Instruct` with the LM-Evaluation-Harness for: MMLU-Pro Wang et al.

---

**Algorithm 1** FlashHead.

---

**Input:** Hidden vector generated by the model body $\mathbf{h} \in \mathbb{R}^d$, where $d$ is the embedding size, Embedding Matrix of all tokens $\mathbf{E} \in \mathbb{R}^{v \times d}$, where $v$ is the total number of tokens in the vocabulary.
**Hyperparameters:** Number of probes $p$, Number of clusters $k$
**Output:** Next token $\mathbf{t} \in \mathbb{R}^d$
**Initialize:** cluster centroids $\mathbf{C} \in \mathbb{R}^{k \times d} \leftarrow$ cluster $\mathbf{E}$ into $k$ clusters
**Inference:**
$\quad \mathbf{c}' \in \mathbb{R}^k \leftarrow \mathbf{C} \times \mathbf{h}$              *(compute centroid logits)*
$\quad \mathbf{C}_p \in \mathbb{R}^{k \times p} \leftarrow$ select $p$ centroids given centroid logits $\mathbf{c}'$
$\quad \tilde{\mathbf{E}} \in \mathbb{R}^{x \times d} \leftarrow$ select tokens from $\mathbf{E}$ corresponding to selected centroids $\mathbf{C}_p$, according to **C2T**.
$\quad \mathbf{z}' \in \mathbb{R}^x \leftarrow \tilde{\mathbf{E}} \times \mathbf{h}$             *(compute token logits)*
$\quad t \in \mathbb{R}^d \leftarrow$ select next token based on token logits.

---

Table 1: Evaluation on LM benchmarks. MMLU-Pro (exact_match, custom-extract), HellaSwag (acc_norm), IFEval (avg. of four metrics), BoolQ (acc_norm), BBH (exact_match), TruthfulQA-gen (bleu_acc), GSM8K (exact_match) for Llama 3.2 1B Instruct as the baseline.

| Method | MMLU Pro | Hella Swag | IFEval | BoolQ | BBH | Truthful QA | GSM8K |
|---|---|---|---|---|---|---|---|
| Baseline | 0.18 | 0.59 | 0.45 | 0.69 | 0.38 | 0.36 | 0.46 |
| Vocab. Trimming | **0.18** | 0.53 | 0.35 | 0.65 | 0.37 | 0.36 | **0.46** |
| SVDSoftmax | 0.16 | 0.44 | 0.44 | 0.69 | 0.13 | 0.36 | 0.26 |
| FGD | — | N/A | 0.32 | N/A | — | **0.42** | — |
| FlashHead | **0.18** | **0.59** | **0.45** | **0.69** | **0.38** | 0.36 | **0.46** |

(2024), HellaSwag Zellers et al. (2019), IFEval Zhou et al. (2023), BoolQ Clark et al. (2019), BBH Suzgun et al. (2022), TruthfulQA-gen Lin et al. (2022b), and GSM8K (8-shot CoT). We use a combination of `alpaca_eval` Taori et al. (2023); Dubois et al. (2023), `MATH-Hard` Hendrycks et al. (2021) and XNLI Conneau et al. (2018) to get a detailed head-level fidelity on challenging multilingual/logic datasets. For probability-based tasks (e.g., HellaSwag, BoolQ), FlashHead uses Monte Carlo sampling to estimate log-likelihoods. To keep runtime tractable, we average over a 100-example subset. We refer the reader to Appendix D.1 for more details on the Monte Carlo sampling.

**Baselines.** We compare FlashHead to training-free baselines: *Vocabulary Trimming* Ushio et al. (2023), *SVDSoftmax* Shim et al. (2017), and *Fast Graph Decoder (FGD)* Zhang et al. (2018). FGD is CPU-only and float32 (as in the original implementation), so it is omitted where GPU or probability-based scoring is required. For comparability, we keep method hyperparameters fixed across tasks: FlashHead uses $c$=8016 clusters, $p$=512 probes (unless varied in ablations or used for other models). Vocabulary Trimming keeps 64,000 tokens (50% of the baseline vocabulary) while SVDSoftmax uses window size 256 (1/8 of embedding) and 12,000 top-$n$, both in line with original implementations. In case of FGD, we use a top $K$ of the nearest neighbors 384, $efSearch = 300$, and $M = 40$, which are significantly larger than the values used in the original paper Zhang et al. (2018) to match the larger vocabulary size for the baseline model.

## 4.1 COMMON EVALUATION BENCHMARKS

Table 1 summarizes task-level performance. FlashHead is the only method to consistently *match the baseline* on all aggregate metrics (within rounding). Notably, FlashHead accomplishes this feat while being significantly faster to execute on-device, see Section 4.3. In Appendix C we also provide the variance across these tasks between independent clustering runs.

Table 2: Top-$k$ Containment ($c$=8016, $p$=512) vs. training-free baselines on `Llama-3.2-1B-Instruct`. For $k$=1, containment = accuracy.

| Method | Top-1 | | | Top-3 | | |
|---|---|---|---|---|---|---|
| | Alpaca | MATH-Hard | XNLI | Alpaca | MATH-Hard | XNLI |
| Baseline | 1.00 | 1.00 | 1.00 | 1.00 | 1.00 | 1.00 |
| Vocab. Trimming | 0.99 | 0.99 | 0.51 | **1.00** | **1.00** | 0.69 |
| SVDSoftmax | 0.94 | 0.96 | 0.96 | 0.99 | 0.99 | 0.99 |
| FlashHead | **1.00** | **1.00** | **0.97** | **1.00** | **1.00** | **1.00** |

Table 3: GPU latency (ms) for head (TPOT$^H$) and full model (TPOT) under bfloat16 and int4 heads. Note that FGD lacks GPU support. Extended CPU and parameter results appear in the Appendix A

| Method | TPOT$^H$ ↓ | TPOT (BF16) ↓ | TPOT (INT4) ↓ |
|---|---|---|---|
| Baseline | 1.94 | 7.69 | 3.60 |
| Vocab. Trimming | 1.07 (1.81×) | 6.82 (1.13×) | 2.73 (1.32×) |
| SVDSoftmax | 0.61 (3.18×) | 6.36 (1.21×) | 2.27 (1.59×) |
| FGD | N/A | N/A | N/A |
| FlashHead | **0.40 (4.85×)** | **6.15 (1.25×)** | **2.06 (1.75×)** |

## 4.2 TOP-K CONTAINMENT

To isolate the classification head, we run both the baseline head and modified ones under identical hidden states and KV-caches and compute *Top-$k$ Containment*: the fraction of cases where the token produced by a method appears in the baseline head's top-$k$. The results for FlashHead and other methods is aggregated in Table 2. FlashHead alone attains a near-perfect match in top-1 (within rounding) on all datasets except for the challenging multilingual dataset XNLI where it achieves 97%. When looking at top-3, FlashHead alone achieves a near-perfect match even for XNLI.

## 4.3 ON-DEVICE LATENCY

We report TPOT$^H$ (head-only) and end-to-end (model-level) TPOT on an NVIDIA RTX 3500 Ada Generation GPU; additional CPU results and effective parameter reductions are deferred to Appendix A due to space. Unless noted otherwise, latencies are measured using vLLM Kwon et al. (2023), averaged over 128 generated tokens, averaged over 100 prompts, with 10 warm-up prompts, in either bfloat16 or W4A16 (int4) precisions for model computations.

Unless otherwise stated, the batch-size is 1. Because our focus is edge-inference and consumer devices, where both memory is limited and user queries must be answered immediately, batching is typically not applicable. As prior work shows Jiang et al. (2018); Wofk et al. (2019); Lin et al. (2022a)), latency and throughput are commonly reported at batch size 1 to reflect realistic on-device conditions.

Table 3 shows that FlashHead achieves the best TPOT$^H$ and significantly improves full-model TPOT.

## 4.4 GENERALIZATION ACROSS MODELS

Table 4 demonstrates that the head-level speedups translate across families (Llama-3.2, Llama-3.1, Qwen-3, Gemma-3). These results highlight FlashHead's robustness as a drop-in head for a wide variety of language models spanning parameter ranges from 270 million parameters to 8 billion.

Table 4: BBH and GPU latency improvements across models. Values in parentheses are speedup factors vs. the dense head. A comprehensive accuracy table and CPU metrics are included in Appendix B. Note that the 8 billion parameter Llama 3.1 model is too big for inference in bfloat16 precision (Out-of-Memory, OOM).

| Model | BBH $\uparrow$ | TPOT$^H$ $\downarrow$ | TPOT (BF16) $\downarrow$ | TPOT (INT4) $\downarrow$ |
|---|---|---|---|---|
| Llama-3.2-1B | $0.38 \rightarrow 0.38$ | $1.94 \rightarrow 0.40$ $(4.9\times)$ | $7.69 \rightarrow 6.15$ $(1.2\times)$ | $3.60 \rightarrow 2.06$ $(1.8\times)$ |
| Llama-3.2-3B | $0.57 \rightarrow 0.57$ | $2.13 \rightarrow 0.68$ $(3.1\times)$ | $18.60 \rightarrow 17.15$ $(1.1\times)$ | $7.11 \rightarrow 5.66$ $(1.3\times)$ |
| Llama-3.1-8B | $0.71 \rightarrow 0.70$ | $2.72 \rightarrow 1.21$ $(2.2\times)$ | OOM | $13.55 \rightarrow 12.04$ $(1.1\times)$ |
| Qwen-3-1.7B | $0.45 \rightarrow 0.45$ | $1.61 \rightarrow 0.45$ $(3.6\times)$ | $9.97 \rightarrow 8.81$ $(1.1\times)$ | $4.85 \rightarrow 3.69$ $(1.3\times)$ |
| Gemma-3-270M | $0.27 \rightarrow 0.27$ | $0.99 \rightarrow 0.37$ $(2.7\times)$ | $2.52 \rightarrow 1.90$ $(1.3\times)$ | $2.38 \rightarrow 1.76$ $(1.4\times)$ |
| Gemma-3-1B | $0.38 \rightarrow 0.38$ | $1.66 \rightarrow 0.52$ $(3.2\times)$ | $6.77 \rightarrow 5.63$ $(1.2\times)$ | $4.12 \rightarrow 2.98$ $(1.4\times)$ |

Table 5: Impact of head quantization on BBH and GPU TPOT$^H$. Latencies are measured and quantization is performed with the transformers Wolf et al. (2020) and HQQ Badri & Shaji (2023) libraries, due to their support of quantizing language model heads.

| Method | Precision | BBH $\uparrow$ | GPU TPOT$^H$ $\downarrow$ |
|---|---|---|---|
| Baseline | BF16 | 0.381 | 1.433 |
| | INT4 | 0.369 | 0.486 |
| FlashHead | BF16 | **0.381** | **0.320** |
| | INT4 | **0.379** | **0.258** |

## 4.5 ABLATIONS

**Quantization of the head.** Quantizing dense LM heads often degrades accuracy, which is why deployment stacks typically avoid it. In contrast, FlashHead's stage-1 (centroid scoring) is robust to low-bit quantization because final token probabilities are re-generated in stage-2 in higher precision. Table 5 shows that FlashHead with an int4 stage-1 achieves TPOT$^H$ gains over a quantized dense head while preserving BBH accuracy much better (bfloat16 FlashHead already matches the baseline).

**Equal-sized clustering.** Equal-size clusters enable a dense cluster-to-token mapping ($C2T$) with predictable memory access and lower gather overhead on accelerators. Table 6 quantifies the benefit: equal-sized clustering simultaneously *improves accuracy and reduces* TPOT$^H$.

**Number of clusters and probes.** FlashHead exposes a smooth latency–accuracy trade-off: increasing probes $p$ or clusters $c$ improves containment, with modest effect on TPOT because cluster size $b=v/c$ shrinks as $c$ grows. Table 7 shows that while these parameters can be used to trade accuracy for further latency gains, we pick ($c=8016, p=512$) as the default to showcase FlashHead's unique ability to maintain accuracy while delivering significant model-level speedups.

## 5 DISCUSSION

FlashHead is most impactful for small and mid-scale language models, where the classification head constitutes a large fraction of parameters and runtime. In such settings, our results indicate that FlashHead reduces the head to a negligible share of overall cost, yielding significant model-level speedups without any fine-tuning.

A promising direction is extending FlashHead to server-side deployment scenarios where large batch sizes are critical. In particular, speculative decoding Zhao et al. (2025) has emerged as a key technique to accelerate inference by generating multiple candidate tokens in parallel. This line of work would adapt FlashHead beyond edge inference toward high-throughput server settings, where latency–throughput trade-offs differ substantially from consumer hardware. Future work may also explore tighter integration with inference frameworks, low-level kernel optimizations, and exten-

Table 6: Equal vs. Unequal clustering on BBH and GPU TPOT$^H$.

| Method | BBH $\uparrow$ | GPU TPOT$^H$ $\downarrow$ |
|---|---|---|
| Unequal clusters | 0.371 | 0.520 |
| Equal clusters | **0.381** | **0.320** |

Table 7: Top-1 containment and GPU latency as a function of clusters $c$ and probes $p$. We report head latency (TPOT$^H$) and full-model TPOT in BF16/INT4.

| #clusters | #probes | Top-1 | | | GPU Latency (ms) | | |
|---|---|---|---|---|---|---|---|
| | | Alpaca | MATH-Hard | XNLI | TPOT$^H$ $\downarrow$ | TPOT (BF16) $\downarrow$ | TPOT (INT4) $\downarrow$ |
| Baseline | | 1.00 | 1.00 | 1.00 | 1.94 | 7.69 | 3.60 |
| 4008 | 128 | 0.99 | 0.99 | 0.93 | 0.18 (10.78×) | 5.93 (1.30×) | 1.84 (1.96×) |
| | 256 | 0.99 | 0.99 | 0.95 | 0.33 (5.88×) | 6.08 (1.26×) | 1.99 (1.80×) |
| | 512 | 1.00 | 1.00 | 0.97 | 0.58 (3.34×) | 6.33 (1.21×) | 2.24 (1.61×) |
| 8016 | 128 | 0.98 | 0.99 | 0.93 | 0.12 (16.17×) | 5.87 (1.31×) | 1.78 (2.02×) |
| | 256 | 0.99 | 1.00 | 0.96 | 0.17 (11.41×) | 5.92 (1.30×) | 1.83 (1.97×) |
| | 512 | 1.00 | 1.00 | 0.97 | 0.40 (4.85×) | 6.15 (1.25×) | 2.06 (1.75×) |
| 16032 | 128 | 0.99 | 1.00 | 0.96 | 0.21 (9.24×) | 5.96 (1.29×) | 1.87 (1.92×) |
| | 256 | 0.99 | 0.99 | 0.97 | 0.28 (6.93×) | 6.03 (1.28×) | 1.94 (1.86×) |
| | 512 | 1.00 | 1.00 | 0.99 | 0.30 (6.47×) | 6.05 (1.27×) | 1.96 (1.84×) |

sions beyond text, such as adapting FlashHead to multimodal models. Another promising direction is to combine FlashHead with training-time modifications, or to accelerate the training process itself, potentially improving efficiency further.

There are also limitations. FlashHead does not provide a closed-form probability distribution over the entire vocabulary. This is not a drawback at inference time, but for evaluation in a research setting, we need to compute probability distributions over all tokens in the vocabulary which we currently do via Monte Carlo simulations, see Appendix D.1 for more details, which slows down likelihood-based evaluation. The clustering itself introduces modest additional storage, though this overhead is negligible compared to full model size.

## 6 CONCLUSIONS

We introduced FlashHead, the first training-free and hardware-friendly replacement for the dense classification head in language models. By combining equal-sized clustering, aggressive multi-probe retrieval, probabilistic probe sampling, and selective quantization, FlashHead collapses the head from a dominant bottleneck into a negligible cost. Across Llama-3.2, Llama 3.1, Gemma-3, and Qwen-3, it preserves accuracy while delivering up to 1.75× end-to-end speedups, even at billion-parameter scales. By removing a key obstacle to deployment on consumer hardware, FlashHead sets a new baseline for output-layer design and opens the path toward faster, smaller, and more widely usable language models.

## 7 REPRODUCIBILITY STATEMENT

We have made all efforts to ensure that the results presented in this paper can be independently reproduced. The FlashHead algorithm is clearly outlined as pseudocode, allowing readers to understand and re-implement it without ambiguity. All hyperparameters and experimental settings are reported in detail in the respective tables to enable faithful reproduction of the experiments. Furthermore, we provide code implementations of FlashHead for several models. We believe these steps will enable the community to readily reproduce, verify, and build upon our work.

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

## A  APPENDIX: EXPANDED ON-DEVICE BENCHMARKS

In the main text (Section 4.3), we presented GPU results for FlashHead and baselines, demonstrating that FlashHead consistently reduces both head-only and full-model latency. Here, we expand those results with additional **CPU latencies** and **active parameter counts** for completeness. The CPU device is Intel Core i7-10870H @ 242 2.20GHz × 16 CPU, and measurements are performed using the transformers Wolf et al. (2020) and HQQ Badri & Shaji (2023) libraries.

These results confirm that FlashHead delivers speedups across both CPU and GPU, while reducing the number of active parameters in the head to a fraction of the dense baseline.

Table 8: On-device latencies and parameter counts for all methods. Latencies are reported in time-per-output-token (ms), averaged over 128 generated tokens. Factors in parentheses indicate relative speedup or reduction compared to the baseline.

(a) GPU Latency

| Method | $\text{TPOT}^H \downarrow$ | TPOT (BF16) $\downarrow$ | TPOT (INT4) $\downarrow$ |
|---|---|---|---|
| Baseline | 1.94 | 7.69 | 3.60 |
| Vocab. Trimming | 1.07 (1.81×) | 6.82 (1.13×) | 2.73 (1.32×) |
| SVDSoftmax | 0.61 (3.18×) | 6.36 (1.21×) | 2.27 (1.59×) |
| FGD | N/A | N/A | N/A |
| FlashHead | **0.40 (4.85×)** | **6.15 (1.25×)** | **2.06 (1.75×)** |

(b) CPU Latency (BF16)

| Method | $\text{TPOT}^H \downarrow$ | TPOT $\downarrow$ |
|---|---|---|
| Baseline | 15.92 | 85.75 |
| Vocab. Trimming | 7.74 (2.06×) | 77.56 (1.11×) |
| SVDSoftmax | 30.94 (0.51×) | 100.77 (0.85×) |
| FGD | 4.74 (3.36×) | 74.56 (1.15×) |
| FlashHead | **3.73 (4.27×)** | **73.55 (1.17×)** |

(c) Active Parameters

| Method | $\text{Params}^H \downarrow$ | Params $\downarrow$ |
|---|---|---|
| Baseline | 263M | 1.236B |
| Vocab. Trimming | 131M | 1.104B |
| SVDSoftmax | 59M | 1.032B |
| FGD | **0.79M** | **974M** |
| FlashHead | 33M | 1.006B |

## B  APPENDIX: EXPANDED RESULTS ACROSS MODELS

In Section 4, we showed that FlashHead generalizes across model families and scales, providing consistent speedups from small (270M) to large (8B) models. Here, we provide a more comprehensive view with two sets of results: (a) benchmark accuracy across LM tasks, and (b) efficiency metrics including head parameters, total parameters, and latency. These expanded results confirm that FlashHead's gains are robust across model families.

Table 9: Evaluation of models across (a) LM benchmarks and (b) efficiency metrics. We exclude BoolQ and HellaSwag due to the high cost of Monte Carlo simulations. Params[H] and Params indicate active and total parameters.

(a) LM evaluation benchmarks

| Model | Method | MMLU-Pro | BBH | Truthful QA | IFEval | GSM8k |
|---|---|---|---|---|---|---|
| Llama-3.2-3B | Baseline | 0.31 | 0.57 | 0.57 | 0.57 | 0.77 |
| | FlashHead | 0.31 | 0.57 | 0.58 | 0.56 | 0.77 |
| Llama-3.1-8B | Baseline | 0.41 | 0.71 | 0.62 | 0.53 | 0.85 |
| | FlashHead | 0.41 | 0.70 | 0.62 | 0.52 | 0.85 |
| Llama-3.2-1B | Baseline | 0.18 | 0.38 | 0.36 | 0.45 | 0.47 |
| | FlashHead | 0.18 | 0.38 | 0.36 | 0.45 | 0.46 |
| Qwen-3-1.7B | Baseline | 0.38 | 0.45 | 0.47 | 0.24 | 0.13 |
| | FlashHead | 0.38 | 0.45 | 0.47 | 0.25 | 0.12 |
| Gemma-3-1B | Baseline | 0.15 | 0.38 | 0.31 | 0.55 | 0.42 |
| | FlashHead | 0.15 | 0.38 | 0.31 | 0.49 | 0.39 |
| Gemma-3-270M | Baseline | 0.09 | 0.27 | 0.31 | 0.32 | 0.02 |
| | FlashHead | 0.09 | 0.27 | 0.32 | 0.30 | 0.02 |

(b) Efficiency and throughput metrics

| Model | Method | Params[H] ↓ | Params ↓ | TPOT[H] ↓ | TPOT (BF16) ↓ | TPOT (INT4) ↓ |
|---|---|---|---|---|---|---|
| Llama-3.2-1B | Baseline | 262M | 1.236B | 1.94 | 7.69 | 3.60 |
| | FlashHead | 33M | 1.006B | 0.40 (4.85×) | 6.15 (1.25×) | 2.06 (1.75×) |
| Llama-3.2-3B | Baseline | 394M | 3.213B | 2.13 | 18.60 | 7.11 |
| | FlashHead | 50M | 2.869B | 0.68 (3.13×) | 17.15 (1.08×) | 5.66 (1.26×) |
| Llama-3.1-8B | Baseline | 525M | 8.030B | 2.72 | OOM | 13.55 |
| | FlashHead | 100M | 7.605B | 1.21 (2.25×) | OOM | 12.04 (1.13×) |
| Qwen-3-1.7B | Baseline | 311M | 1.721B | 1.61 | 9.97 | 4.85 |
| | FlashHead | 36M | 1.446B | 0.45 (3.58×) | 8.81 (1.13×) | 3.69 (1.31×) |
| Gemma-3-270M | Baseline | 168M | 268M | 0.99 | 2.52 | 2.38 |
| | FlashHead | 21M | 121M | 0.37 (2.68×) | 1.90 (1.33×) | 1.76 (1.35×) |
| Gemma-3-1B | Baseline | 302M | 1.000B | 1.66 | 6.77 | 4.12 |
| | FlashHead | 38M | 736M | 0.52 (3.19×) | 5.63 (1.20×) | 2.98 (1.38×) |

## C ROBUSTNESS TO INDEPENDENT CLUSTERING RUNS

FlashHead relies on a one-time offline spherical $k$-means clustering of token embeddings. Because $k$-means is sensitive to random initializations, it is natural to ask whether rebuilding the clusters from scratch affects either predictive quality.

To answer this question, we repeat the common evaluation benchmarks used for the main FlashHead results reported in the main text *five (5) times*, regenerating the complete set of $8\,016$ clusters in each run.

Table 10 reports the mean and standard deviation across the five runs. Rebuilding the clustering neither helps nor hurts any benchmark.

Table 10: Mean and standard deviation of FlashHead performance on LM-Eval benchmarks (Llama-3.2-1B-Instruct); scores are averaged over five independent clusterings.

| MMLU-Pro | BBH | TruthfulQA (gen) | IFEval | GSM8K (chat) |
|---|---|---|---|---|
| $0.181 \pm 0.001$ | $0.377 \pm 0.004$ | $0.363 \pm 0.002$ | $0.452 \pm 0.003$ | $0.465 \pm 0.003$ |

## D  ESTIMATING THE MARGINAL DISTRIBUTION

For evaluation tasks such as HellaSwag and BoolQ, we need log-likelihoods w.r.t. the *full* vocabulary, but enumerating all $c$ clusters is precisely what FlashHead avoids.

FlashHead first samples a *subset* of centroids $S \subseteq \mathcal{C} := \{\mathcal{C}_1, \ldots, \mathcal{C}_c\}$ (size $n_{\text{probes}}$) and then samples a token $t$ within those centroids, see Figure 1. Consequently the exact marginal $p(t \mid \mathbf{h}) = \mathbb{E}_S\big[p(t \mid \mathbf{h}, S)\big]$ requires an expectation over $\binom{\mathcal{C}}{n_{\text{probes}}}$ subsets, which is infeasible to compute.

For results of FlashHead reported in the main text where the full log-likelihood is required, we estimate the true underlying distribution using a Monte-Carlo approximation, which we explain in more detail in Appendix D.1.

All hyper-parameters (number of clusters, number of probes, vocabulary size, window size, etc.) are identical to those listed in Section 4 of the main text. Datasets and their settings used in this section are the same as those originally described in Section 4.1. To ensure good precision, we accumulate probabilities in 64-bit floating point format (everything else remains in bfloat16).

### D.1  MONTE-CARLO SIMULATION

For each of a total of $N$ probe sets $S_1, \ldots, S_N \sim p(S \mid \mathbf{h})$ we evaluate the full conditional distribution $p(v \mid \mathbf{h}, S_i)$ and accumulate it

$$\hat{s}_N(v \mid \mathbf{h}) \;=\; \sum_{i=1}^{N} p(v \mid \mathbf{h}, S_i).$$

Averaging these $N$ conditional distributions yields a properly normalised marginal,

$$\hat{p}_N(v \mid \mathbf{h}) \;=\; \frac{\hat{s}_N(v \mid \mathbf{h})}{N} \;=\; \frac{1}{N} \sum_{i=1}^{N} p(v \mid \mathbf{h}, S_i), \qquad \sum_{t \in \mathcal{V}} \hat{p}_N(v \mid \mathbf{h}) = 1.$$

Because each probe set contributes a full distribution rather than a single token, this Monte-Carlo estimator has substantially lower variance than sampling individual tokens.

If a token is never present in any of the $S_i$, then $\hat{p}_N(v \mid \mathbf{h}) = 0$ and the log-likelihood becomes $-\infty$. To avoid undefined accuracies we clip such zero entries to the minimum non-zero probability observed in the same context before computing task metrics.

Figure 2 shows the accuracy obtained on BoolQ and HellaSwag as a function of $N$. Beyond $N \approx$ 10,000 the curves are flat, so we used that setting for all headline results. With smaller sample sizes ($< 10{,}000$) the "never-sampled-token" issue becomes more common. All Monte-Carlo runs were performed on an NVIDIA RTX 3500 Ada Geneneration; generating the 10 000 probe sets for each evaluation subset of HellaSwag or BoolQ took roughly 2 seconds.

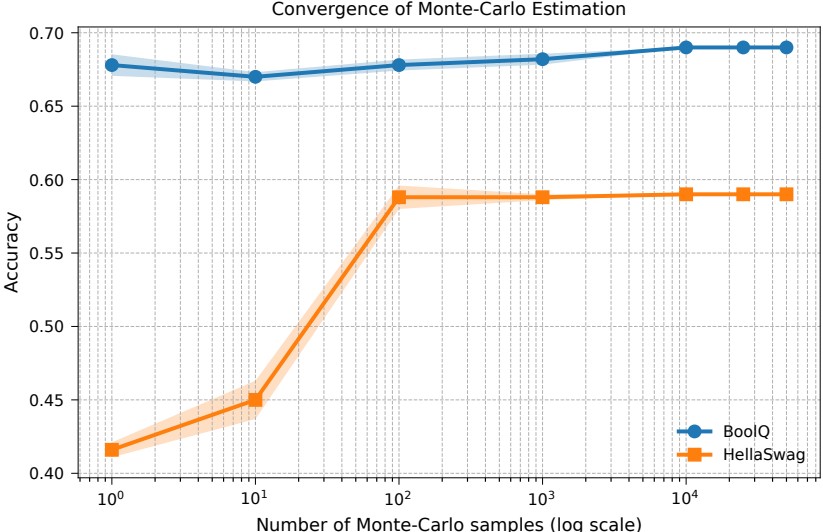

Figure 2: Convergence of the Monte-Carlo estimator when used to evaluate accuracy on the BoolQ and HellaSwag datasets. An average is taken over five independent clustering procedures with different random seeds. Shaded ribbons show $\pm 1$ s.e.

# E  MODELS, HYPER-PARAMETERS, DATASETS AND EVALUATION SETTINGS

In our experiments, we explore all methods using the baseline Llama-3.2-1B-Instruct model. The alternative heads are initialized with their own set of hyper-parameters (e.g., number of clusters, vocabulary size). We list all such settings in Table 11.

To evaluate model performance (LM-Eval benchmark tasks and Top-K Containment) and on-device latencies, we use several datasets listed in Table 12. For each dataset we provide both a formal literature reference and direct links to their public HuggingFace or GitHub repositories. In case of latency and Top-K Containment benchmarks, we use the Alpaca (eval) dataset and the test splits for the MATH-Hard and XNLI datasets. In the case of vocabulary trimming, we use the train split to mine the high frequency tokens to use for the reduced vocabulary.

All LM-Eval benchmarks are conducted through the LM-Evaluation-Harness framework. Table 13 details, for each benchmark, the reported metric and the relevant generation settings.

# F  EXAMPLES OF CLUSTERS

To demonstrate the clustering used by FlashHead, we provide ten (out of the total 8,016) randomly selected clusters in Table 14. These examples illustrate how the tokenizer groups related spellings, capitalization variants, and domain-specific affixes.

Table 11: Methods and hyper-parameters used in the various implementations.

| Method | Hyper-parameters |
|--------|------------------|
| **Baseline (LLaMA-3.2 1B Instruct)** | num_hidden_layers: 16 |
| | hidden_size: 2048 |
| | intermediate_size: 8192 |
| | num_attention_heads: 32 |
| | num_key_value_heads: 8 |
| | head_dim: 64 |
| | hidden_act: silu |
| | mlp_bias: false |
| | attention_bias: false |
| | rms_norm_eps: 1e-5 |
| | max_position_embeddings: 131072 |
| | rope_scaling: {factor: 32.0, high_freq: 4.0 |
| |     low_freq: 1.0, original_max: 8192} |
| | rope_theta: 500000.0 |
| | bos_token_id: 128000 |
| | eos_token_id: [128001, 128008, 128009] |
| | vocab_size: 128256 |
| | dtype: bfloat16 |
| **Vocab. Trimming** | vocab_size: 64,000 |
| | Top tokens filtered by frequency based on the Alpaca (train) dataset for all LM-Eval tasks, and the relevant train dataset for Top-K Containment evaluations |
| **SVDSoftmax** | window: 256 |
| | top_n: 12,000 |
| **Spherical K-Means** | n_clusters: 100, 20. |
| **Fast Graph Decoder (FGD)** | K: 384 |
| | ef_search: 300 |
| | index_M: 40 |
| | ef_construction: 300 |
| | FGD implementation is CPU-only with dtype: float32 |

Table 12: Overview of the datasets used in this paper.

| Dataset name | Reference | Link |
|--------------|-----------|------|
| Alpaca (train and eval) | Taori et al. (2023); Dubois et al. (2023) | tatsu-lab/alpaca, tatsu-lab/alpaca_eval |
| MATH-Hard | Hendrycks et al. (2021) | lighteval/MATH-Hard |
| XNLI | Conneau et al. (2018) | facebook/xnli |
| MMLU-Pro | Wang et al. (2024) | TIGER-Lab/MMLU-Pro |
| Hella-Swag | Zellers et al. (2019) | hellaswag |
| IFEval | Zhou et al. (2023) | google/IFEval |
| BoolQ | Clark et al. (2019) | google/boolq |
| BBH | Suzgun et al. (2022) | suzgunmirac/BIG-Bench-Hard |
| Truthful-QA (gen) | Lin et al. (2022b) | sylinrl/TruthfulQA |
| OBQA | Mihaylov et al. (2018) | allenai/openbookqa |

Table 13: Evaluation settings for LM-Eval tasks.

| Benchmark | Reported metric | Settings |
|---|---|---|
| MMLU-Pro | exact_match,custom-extract | max_gen_toks: 2048 num_fewshot: 5 |
| HellaSwag | acc_norm | num_fewshot: 0 |
| IFEval | average of (prompt_strict, instruction_strict, prompt_loose, instruction_loose acc) | num_fewshot: 0 |
| | | max_gen_toks: 1280 |
| BoolQ | acc | num_fewshot: 0 |
| BBH | exact_match,get_answer | num_fewshot: 3 max_gen_toks: 1024 |
| Truthful-QA (gen) | bleu_acc | num_fewshot: 0 |
| OpenBookQA (OBQA) | acc_norm | num_fewshot: 0 |

Table 14: Tokens for 10 randomly sampled clusters from LLaMA-3.2-1B-Instruct. Tokens are comma separated without extra padding; perceived leading spaces belong to the tokens themselves.

| Cluster | Tokens |
|---|---|
| 661 | change, change, Change, Change, _change, .change, _CHANGE, CHANGE, CHANGE, Changing, -change, changing, (change, Changing, \tchange, /change |
| 1593 | javax, .junit, javafx, TestBed, .slf, .jupiter, .testng, .jboss, .joda, NUnit, JUnit, .hamcrest, \TestCase, /gtest, junit, .assertj |
| 3824 | distr, district, distur, districts, disturbing, distress, disturb, distortion, distraction, disturbed, distorted, disturbance, distort, distressed, _district, disturbances |
| 4336 | UART, USB, usb, _UART, _USB, PWM, USART, uart, _PWM, _usb, UART, _uart, pwm, _USART, _pwm, PWM |
| 4389 | exem, exempt, exemption, nesty, exponential, pardon, exemptions, exo, Amnesty, spree, ambush, amnesty, pard, Ames, impunity, exon |
| 4584 | done, done, Done, dice, _done, Done, .done, (done, \tdone, _DONE, DONE, Gone, .Done, DONE, undone, -done |
| 5065 | ictureBox, .ToolStrip, ToolStrip, .toolStrip, pictureBox, .menuStrip, toolStrip, .ToolStripButton, pictureBox, ToolStrip, .toolStripSeparator, toolStrip, .toolStripButton, Bunifu, PictureBox, PictureBox |
| 5251 | pack, pack, Pack, PACK, .pack, _PACK, packs, Pack, _pack, PACK, pak, Packers, -pack, -Pack, Packs, (pack |
| 7376 | peed, acceler, acceleration, hast, accelerate, accel, Acceler, accelerated, _accel, Acceler, accelerator, accel, accelerating, .accel, Acceleration, accelerometer |
| 7656 | check, check, Check, Check, _check, .check, checks, checking, CHECK, -check, (check, Checks, \tcheck, Checking, Checking, checking |