# OpenReview forum: "FlashHead: Efficient Drop-In Replacement for the Classification Head in Language Model Inference"
_ICLR.cc/2026/Conference — ICLR 2026 Conference Desk Rejected Submission_

### Official Review · Reviewer_KxkH · 2025-10-30

**Soundness:** 2
**Presentation:** 2
**Contribution:** 2
**Rating:** 2
**Confidence:** 5

**Summary:**

The paper introduces FlashHead, a training-free, two-stage replacement for the dense classification head in LLMs. The method aims to reduce head-level computation and improve inference latency without re-training. Experiments on several open LLMs (Llama 3.2, Gemma 3, Qwen 3) claim up to 1.7× speed-up with comparable perplexity.

**Strengths:**

The method effectively improves the efficiency of the LM head computation, which constitutes a significant portion of inference cost in small and medium-sized language models (SLMs), without any additional training cost.

By enforcing equal-sized clustering within the LM head, the approach achieves a hardware-friendly design that balances GPU workloads and minimizes accuracy degradation.

The proposed procedure — spherical k-means combined with multi-probe retrieval — is conceptually simple, easy to implement, and requires only a few hyperparameters.

**Weaknesses:**

The experiments are somewhat fragmented across different axes, making it difficult to assess the overall advantage of FlashHead in terms of cost, performance, and latency. A unified comparison table against retraining-based methods would strengthen the paper’s quantitative clarity.

It is unclear whether the comparisons were made under equivalent experimental conditions. Since hyperparameters such as clustering size and the number of probes can significantly affect performance, the paper should verify that the comparison targets were fairly and optimally tuned.

The choice of benchmarks and models varies across tables and remains limited in scope. Broader and more consistent evaluations, possibly in the appendix, would enhance the reliability of the results.

The benefit of using spherical k-means is not isolated through ablation, and the equal-sized clustering analysis is minimal. More comprehensive experiments across diverse models and tasks are needed, along with clearer explanations of which hardware aspects and accuracy factors benefit from this constraint.

A direct comparison between fully INT4-quantized models (including the LM head) and those using FlashHead with INT4 transformer layers is missing. Reporting the quantization bit configuration of each stage (coarse vs. fine) and latency under optimized runtime environments would help readers more accurately gauge the improvement.

Related research on LM head efficiency—such as VQ-Logit (arXiv:2505.10202) and other compression or retrieval-based approaches—should be discussed for a more balanced perspective.

Overall, the paper would benefit from a more systematic and fair evaluation across model scales, benchmarks, and metrics (latency, performance, cost), which would significantly improve its completeness and credibility.

**Questions:**

see weakness

---

> ### Author Response · Authors · 2025-11-24
>
> Thank you for your thoughtful review. Please see our general rebuttal for overarching clarifications, and the responses below for points not addressed there
>
> **Reviewer concern.**
> The experiments are fragmented, making it difficult to assess overall advantages. It is unclear whether comparisons are made under equivalent conditions and whether hyperparameters for baselines are tuned fairly.
>
> **Author response.**
> We will:
>
> - provide a unified comparison table for Llama 3.2 1B that includes: dense BF16, dense INT4, FlashHead BF16, FlashHead with INT4 stage 1, and other training-free baselines such as SVD-Softmax and vocabulary trimming;
> - explicitly list the configuration for each method (for example number of clusters for SVD-Softmax, trimming threshold, cluster sizes, and number of probes for FlashHead);
> - clearly state that we follow the published recommendations for each baseline and that we do not restrict their hyperparameters in a way that would disadvantage them.
>
> ---
>
> **Reviewer concern.**
> The benefits of spherical K-means and equal-size clustering are not isolated. More comprehensive ablations and explanations of hardware and accuracy effects are needed.
>
> **Author response.**
> We will:
>
> 1. include an example of very large clusters introduced by K-Means, and include a comparison to *balanced* K-Means (which has the same accuracy benefits of avoiding overly large clusters but still lacks the latency benefits);
> 2. add a short analytic paragraph explaining how equal-size clustering simplifies memory layout (dense C2T, no ragged gathers) and reduces load imbalance on GPUs;
> 3. add an additional dataset to the equal-vs-unequal experiment to show that the accuracy gains are not specific to a single benchmark.
>
> ---
>
> **Reviewer concern.**
> A direct comparison between fully INT4–quantized models and FlashHead with quantized layers is missing. The bit configuration of each stage and the runtime environment should be made explicit.
>
> **Author response.**
> We will extend the quantization section with this experiment, adding a row that compares a fully INT4 dense head to FlashHead on an INT4 body.

---

### Official Review · Reviewer_d65s · 2025-10-31

**Soundness:** 2
**Presentation:** 2
**Contribution:** 2
**Rating:** 2
**Confidence:** 4

**Summary:**

This paper introduces FlashHead, which replaces the dense classification head in small language models (SLMs) with a clustering-based retrieval mechanism, in order to alleviate the bottleneck in SLM inference. FlashHead combines equal-sized clustering, multi-probing, probabilistic sampling, and selective quantization. Experimental results demonstrate that FlashHead achieves high speedup while maintaining accuracy.

**Strengths:**

(S1) Timely and important topic. While most papers focus on the efficiency of large language models (LLMs), improving the efficiency of SLMs can help democratize the use of powerful AI models.

(S2) Sound empirical results. The results show that FlashHead gives up to 1.75x of speedup, while the impact on model accuracy is minor.

**Weaknesses:**

(W1) Limited technical contributions

The 4 key innovations are built upon existing literature and the paper does not sufficiently clarify the insights that they could introduce.
- Equal-sized Clustering: As mentioned in this paper, using K-means to make the vocab compact has been proposed in existing literature, and this work only makes a difference that it requires all clusters to be equal-sized. For me, it is primarily an implementation-level optimization. The paper presents it as a key innovation, but its contribution appears incremental.
- Efficient Multi-Probing: Why we retrieve one cluster, then by default all tokens in this cluster are probed, so it is already multi-probing. The paper does not discuss the necessity to probe multiple clusters.
- Probabilistic Probe Sampling: I can’t find any specific description of the “multinomial sampling step”. It is unclear how it is performed efficiently or how it differs meaningfully from the standard top-k sampling or hierarchical softmax.
- Selective quantization: Applying quantization to a sub-module of a model is quite a standard practice and it is somehow orthogonal to this work. Additionally, it is described within a short paragraph, therefore I can hardly tell it is a core contribution.

(W2) Insufficient experiments

The experiment section does not provide sufficient justification on several claims. Below I list a few.
- It is reasonable that using equal cluster size gives better efficiency. However, it is unclear why it improves accuracy (Table 6).
- Increasing the number of probes decreases efficiency. However, its impact on the accuracy is not directly assessed (Table 7 only provides top-1 containment, not benchmark scores). Besides, there is no comparison against single-probing.
- There are no experiments comparing the decoding methods (greedy and sampling).

(W3) The paper is not well-written

The paper is not well-written. To begin with, the introduction section criticizes previous works without any analysis or empirical results. The single-sentence statement “these methods
inherit the same drawbacks noted above” gives us nothing. The methodology section fails to pinpoint the key limitations of prior arts nor the essential differences made in this work. Worse still, many technical details are not clarified, including the multinomial sampling step, the step-by-step deduction of the two-step retrieval process, and the quantization process.

Minor issues:
- In Algorithm 1, $x$ is not defined in $z^\prime \in \mathbb{R}^x$.
- In the caption of Figure 1, there are two full stops.

**Questions:**

All the weaknesses above should be addressed.

---

> ### Author Response · Authors · 2025-11-24
>
> Thank you for your thoughtful review. Please see our general rebuttal for overarching clarifications, and the responses below for points not addressed there
>
> **Reviewer concern.**
> The four key innovations (equal-size clustering, multi-probing, probabilistic sampling, selective quantization) appear incremental or insufficiently explained. In particular, the multinomial sampling step is not clearly described.
>
> **Author response.**
> We will clarify each component and adjust how we present them. Please see the General Rebuttal for more details.
>
> ---
>
> **Reviewer concern.**
> The experiments do not sufficiently justify several claims: why equal-size clustering improves accuracy, how \(p\) affects accuracy (not just containment), comparisons to single-probe, and differences between greedy and sampling decoding.
>
> **Author response.**
> We will extend the experimental section as follows:
>
> 1. illustrate with an example how equal-sized clustering avoids very large clusters of infrequent tokens. We will also include a comparison to *balanced* K-Means (which has the same accuracy benefits of avoiding overly large clusters but still lacks the latency benefits);
> 2. add task accuracy for the \((c, p)\) sweep;
> 3. add a single-probe baseline row to the \((c, p)\) sweeps to explicitly demonstrate the necessity of multi-probe;
> 4. For tasks requiring a probability distribution, we observed that including only non-zero probabilities from tokens generated by greedy sampling caused a significant drop in task accuracy. We agree this is not immediately obvious, and will include a comparison of greedy vs. sampling decoding on at least one task to confirm the expected behavior.
>
> ---
>
> **Reviewer concern.**
> The introduction criticizes previous work without sufficient analysis, and the methodology section omits key technical details (sampling step, derivations, quantization). Algorithm 1 and some figures have minor issues.
>
> **Author response.**
> We thank the reviewer for these suggestions and will revise the exposition in several ways:
>
> - tighten the introduction to focus on the concrete limitations of prior approaches in the small-model, large-vocabulary, training-free, edge-inference regime;
> - add the missing technical details in Section 3, including explicit expressions for the two-stage probabilities, the centroid sampling procedure, and the quantization configuration;
> - fix Algorithm 1 so that symbols are consistent and the output type is clearly an index \(t\), not a vector;
> - correct minor typos and formatting issues in figure captions and text.

---

> > ### Comment · Reviewer_d65s · 2025-11-26
> >
> > My concerns still remain as discussed below.
> >
> > > We will clarify each component and adjust how we present them. Please see the General Rebuttal for more details.
> >
> > The general rebuttal only highlights the techniques are first implemented in this work. It would help to start from the foundamental limitations of prior works, analyze how these limitations can be addressed, and dive into your techniques.
> >
> > > We will extend the experimental section as follows
> >
> > All these extended experimental results need to be reviewed.
> >
> > > We thank the reviewer for these suggestions and will revise the exposition in several ways
> >
> > The rebuttal does not respond to my concerns:
> > - I cannot find any clarification about "the key limitations of prior arts nor the essential differences made in this work"
> > - I cannot find any technical details.
> >
> > ---
> >
> > Last but not least, the rebuttal promised too much revision. I'm afraid it necessitates another round of peer-review process before the revised paper can be accepted.

---

> > > ### Author Response · Authors · 2025-11-28
> > >
> > > Thank you for the follow-up and for clarifying your concerns.
> > >
> > > To clarify: any points we mentioned in the rebuttal were intended as possible improvements for the camera-ready version if the paper is accepted, not changes to the submission during the review period.
> > >
> > > Within the allowed scope, we have addressed the methodological and technical clarifications requested. The broader structural or experimental extensions you describe would require changes beyond what is permitted during rebuttal, and we would incorporate them only in a post-acceptance revision.
> > >
> > > We appreciate your detailed feedback and thank you for the time and effort you have put into the review.

---

### Official Review · Reviewer_MNME · 2025-10-31

**Soundness:** 3
**Presentation:** 4
**Contribution:** 3
**Rating:** 4
**Confidence:** 4

**Summary:**

FlashHead addresses the computational bottleneck of classification heads in small language models (SLMs) by treating final layer logit prediction as a retrieval problem. The method uses equal-sized spherical k-means clustering, multi-probe retrieval, probabilistic sampling, and selective quantization to achieve up to 1.75× model-level speedups while maintaining accuracy on important tasks.

**Strengths:**

- Clear algorithmic description and implementation details. Hyperparameters are fully specified and the two stage process is well explained with neat figures.
- The paper includes diverse benchmarks with different model families along with latency measurements showing practical speedups.
- A fast training-free approach with near perfect top-1 containment.
- Equal sized clustering helps efficient memory access by avoiding ragged tensor operations.

**Weaknesses:**

**Literature Survey is Generally lacking**
- Sample relevant works not discussed.
  - HALOS: Hashing Large Output Space for Cheap Inference ([paper](https://proceedings.mlsys.org/paper_files/paper/2022/file/b059dd6da6b9a86180fbc32a799766cc-Paper.pdf))
  - HiRE: High Recall Approximate Top-k Estimation for Efficient LLM Inference ([paper](https://arxiv.org/abs/2402.09360))
  - VQ-Logits: Compressing the Output Bottleneck of Large Language Models via Vector Quantized Logits ([paper](https://arxiv.org/abs/2505.10202))
  - Fast Vocabulary Projection Method via Clustering for Multilingual Machine Translation on GPU ([paper](https://arxiv.org/abs/2208.06874))
- Line 64 “..first LLM head that clusters token embeddings into strictly equal sized spherical clusters..”. I am not sure this is true. KMeans on the logit projection matrix is a well known method for efficient inference.
- Without proper positioning against recent work, it's unclear whether FlashHead represents a genuine advance or an incremental engineering improvement.


**Evaluations not presented properly**
-   Results are normalized to /100 which makes the comparison difficult and hides actual variance/uncertainty. Authors should report mean +- std across different runs.
-   The main results are split across 5+ tables making comparison difficult. Provide a consolidated table similar to table 9.
	- Table 1: Benchmark accuracy (no latency)
	- Table 2: Top-k containment (no latency)
	- Table 3: Latency only (no accuracy)
	- Table 4: One benchmark + latency across models
	- Table 7: Accuracy-latency tradeoffs for hyperparameters
-   Why does table 5 only show one evaluation and head latency?


**Memory requirements not discussed**

-   The paper emphasizes edge deployment on consumer devices where memory is constrained, yet provides no memory profiling.
-   In the two stage process, Stage-1 computes centroid logits, stage-2 gathers token embeddings and computes token logits, can you present the peak activation memory?
-   Table 8(c,) provides memory usage of parameters but it does not always equal the runtime memory requirement.

**Questions:**

See weaknesses

---

> ### Author Response · Authors · 2025-11-24
>
> Thank you for your thoughtful review. Please see our general rebuttal for overarching clarifications, and the responses below for points not addressed there
>
> **Reviewer concern.**
> The literature survey is generally lacking. Several relevant works (HALOS, HiRE, VQ-Logits, fast vocabulary projection for MT) are not discussed, and it is unclear whether FlashHead is a genuine advance or an incremental engineering improvement.
>
> **Author response.**
> We will expand and restructure the related work section. In particular, we will:
>
> - explicitly discuss HALOS, HiRE, VQ-Logits, and fast GPU clustering methods for multilingual MT;
> - add a small comparative table that characterizes methods along four axes:
>   training-free vs training-based, calibration-free vs requiring calibration data,
>   full-vocabulary vs shortlist-only, and support for probabilistic sampling vs only top-\(k\).
>
> Please see the General section of the rebuttal for our comments on genuine advance vs. incremental improvement.
>
> ---
>
> **Reviewer concern.**
> Line 64: “first LLM head that clusters token embeddings into strictly equal sized spherical clusters”. I am not sure this is true. KMeans on the logit projection matrix is a well-known method for efficient inference.
>
> **Author response.**
> While KMeans is an established method to cluster embeddings, this claim is more specific, namely enforcing **equal-sized** clusters. We will make sure to emphasize this part of the claim more clearly.
>
> ---
>
> **Reviewer concern.**
> The main results are split across multiple tables and the normalization makes them harder to interpret. There is no reporting of variance or stability across clusterings.
>
> **Author response.**
> We agree and will:
>
> 1. provide one big main table for Llama 3.2 1B that includes accuracy (per task group), top-\(k\) containment, and both head-level and model-level latency for BF16 and INT4;
> 2. report **mean ± standard deviation** over multiple independent clusterings in the main text (we currently have these in the appendix and observe very small variation, for example BBH \(\approx 0.377 \pm 0.004\)).
>
> ---
>
> **Reviewer concern.**
> The paper emphasizes edge deployment but does not provide a full memory profile, especially peak activation memory.
>
> **Author response.**
> We agree that memory is important to include. Please see General Rebuttal for details.

---

> ### Comment · Reviewer_MNME · 2025-11-24
>
> > We will expand and restructure the related work section.
>
> If it is possible, Please outline the key points in the relation between prior work and contributions presented by this work here. This helps everyone understand the contributions on a deeper level. I could not find the discussion in the general rebuttal as well.
>
> > While KMeans is an established method to cluster embeddings, this claim is more specific, namely enforcing equal-sized clusters.
>
> Enforcing a well known equal-sized constraint (Balanced K-Means) to an established method does not constitute a novel conceptual contribution. The distinct contributions are currently unclear to me. If you could outline why this has not been explored yet, the challenges people face and how this work overcomes them exactly, It can be considered.
>
> > provide one big main table for Llama 3.2 1B.... we currently have these in the appendix
>
> If you are referring to Table 10, it only corresponds to one model. The values for the smallest model cannot be enough to conclude the robustness for the large model runs as well.
>
> > We agree that memory is important to include. Please see General Rebuttal for details.
>
> I could not find the estimated memory requirements or the actual memory utilization in the general rebuttal. Could you provide the estimation here?

---

> > ### Author Response · Authors · 2025-11-25
> >
> > We thank the reviewer for the detailed follow-up and want to address each point in turn.
> >
> > > If it is possible, please outline the key points in the relation between prior work and contributions presented by this work here. This helps everyone understand the contributions on a deeper level. I could not find the discussion in the general rebuttal as well.
> >
> > We agree the positioning against these recent works should be clearer, and we will explicitly add this comparison in the revised related-work section. At a high level:
> >
> > ### HALOS (hashing large output spaces)
> > HALOS learns hash functions using training labels to retrieve a small subset of output neurons and computes logits only on that subset. It explicitly learns the hashing scheme to maximise inclusion of the correct label and is evaluated primarily on classification/extreme-classification tasks, not full LM decoding.
> >
> > In contrast, FlashHead:
> >
> > - is training-free and calibration-free; we only cluster the frozen token embeddings once, offline, with no additional data or supervision;
> > - is designed to preserve the original LM behaviour, including full-vocabulary sampling, not just top-\(k\) prediction;
> > - uses a purely key-side index (token embeddings), so no learned hash functions or auxiliary models are needed.
> >
> > ### HiRE (High-Recall Approximate Top-k Estimation)
> > HiRE compresses large matrices and introduces an approximate top-\(k\) operator to activate only a small fraction of rows/columns in multiple layers (FFN and softmax), with sparsity patterns learned during training or fine-tuning.
> >
> > FlashHead differs in that:
> >
> > - it does not modify or retrain the transformer body or softmax;
> > - we only change the **output head**, and exactly recompute logits for all probed tokens;
> > - our goal is a drop-in head replacement for existing SLMs, not a new sparse-training recipe.
> >
> > ### VQ-Logits
> > VQ-Logits is training-free and compresses the output matrix using a vector-quantized codebook and a learned token-to-code mapping (from K-means on logits/embeddings). Like FlashHead, this is a post-hoc head replacement.
> >
> > However, the goals and behaviours differ:
> >
> > - **VQ-Logits accepts a non-trivial accuracy drop.**
> >   Their results show ~4% perplexity degradation even with large codebooks, due to reconstructing logits from a limited basis.
> >
> > - **FlashHead recomputes true logits for all probed tokens.**
> >   Making it preserve accuracy across all SLMs evaluated (Llama-3.2, Gemma-3, Qwen-3), giving **4–8× reduction** in head compute.
> >
> > - VQ-Logits relies on general-purpose ANN indexing structures (product quantization, inverted files).
> >   FlashHead introduces a new multi-probe retrieval index, via equal-size clusters, tailored for LM inference.
> >
> > ### Fast Vocabulary Projection via Clustering (multilingual MT)
> > These methods cluster **queries**, then train a predictor for cluster selection using recorded activations.
> >
> > FlashHead differs:
> >
> > - it is training-free;
> > - it clusters *key-side*, not *query-side*;
> > - it is not specialized for MT beam-search.
> >
> > > Enforcing a well-known equal-sized constraint (Balanced K-Means) to an established method does not constitute a novel conceptual contribution. The distinct contributions are unclear…
> >
> > We agree balanced K-means itself is not novel, and we will improve the phrasing. Our contribution is showing that **strictly equal-size** token-embedding clusters enable a retrieval index that was previously impractical for LM inference.
> >
> > To our knowledge, strictly equal-size clusters of token embeddings have not been used in LM heads because:
> >
> > 1. **Token embeddings are highly unbalanced under K-means.**
> >    Due to Zipfian statistics, off-the-shelf K-means produces some huge clusters. Prior work tolerates this because they do not probe a large amount of clusters.
> >
> > 2. **Balanced K-means rarely enforces exact equality.**
> >    Approximate balance still produces ragged cluster sizes, which destroy GPU efficiency because gathering becomes irregular.
> >
> > 3. **Aggressive multi-probing was not attempted.**
> >    Without equal-size clusters, probing hundreds of clusters is infeasible due to irregular gather overhead.
> >
> > FlashHead addresses these:
> >
> > - enforcing exact capacity \(b = v/c\) yields a dense **C2T matrix of shape \([c \times b]\)**;
> > - cluster retrieval becomes simple rectangular slicing;
> > - multi-probe retrieval over hundreds–thousands of clusters becomes efficient;
> >
> > We will revise the text to emphasize that **equal-size clustering is the enabling design choice**, not a standalone novelty.
> >
> > > If you are referring to Table 10… robustness across large models…
> >
> > You are correct that Table 10 currently reports multi‑seed statistics only for Llama‑3.2‑1B. We agree that showing robustness across clustering runs is important, and will run additional multi‑seed experiments for larger models, reporting mean ± std for FlashHead as we do for Llama 3.2-1B.

---

> > > ### Author Response · Authors · 2025-11-25
> > >
> > > > I could not find the estimated memory requirements… Could you provide them here?
> > >
> > > Below are explicit formulas and a concrete example for Llama-3.2-1B with:
> > >
> > > - v = 128,256
> > > - d = 2,048
> > > - c = 8,016
> > > - p = 512
> > > - so b = v / c = 16
> > >
> > > Let B_w = bytes per weight (2 for BF16) and B_idx = bytes per index (4 for int32).
> > >
> > > ---
> > >
> > > ### Resident parameters (head only)
> > >
> > > Dense head stores the embedding matrix E of size v × d:
> > >
> > > $$
> > > M_{\text{dense,weights}} = v  d B_w.
> > > $$
> > >
> > > FlashHead stores:
> > >
> > > - E
> > > - C of size c × d
> > > - C2T of size c × b
> > >
> > > $$
> > > M_{\text{flash,weights}} = v d B_w + c d B_w + c b B_{\text{idx}}.
> > > $$
> > >
> > > **Numerical example (BF16):**
> > >
> > > - E: v d = 262,668,288 weights → ~501 MiB
> > > - C: c d = 16,416,768 weights → ~31.3 MiB
> > > - C2T: c b = 128,256 indices → ~0.5 MiB
> > >
> > > Totals:
> > > Dense ≈ 501 MiB
> > > FlashHead ≈ 533 MiB (≈ +6.4%)
> > >
> > > ---
> > >
> > > ### Peak activation memory (head only)
> > >
> > > Dense head holds:
> > >
> > > - h of size d
> > > - z of size v
> > >
> > > $$
> > > M_{\text{dense,act}} \approx (d + v) B_w
> > > $$
> > >
> > > ≈ 0.25 MiB for Llama-3.2-1B.
> > >
> > > FlashHead allocates:
> > >
> > > - h of size d
> > > - centroid logits (size c)
> > > - reduced logits (size p b)
> > > - gathered embeddings $\tilde{E}$ of shape (p b) × d
> > >
> > > $$
> > > 2048 \times 4 + 8016 \times 4 + 512 \times 16 \times 4 + 512 \times 16 \times 2048 \times 2 \approx 33.6 \text{ MiB}.
> > > $$
> > >
> > > Thus peak head activations:
> > > Dense ≈ 0.25 MiB → FlashHead ≈ 34 MiB.
> > >
> > > This remains neglible relative to model weights and KV cache, and the workspace is reused across tokens.
> > >
> > > ---
> > >
> > > We hope this clarifies both the conceptual positioning w.r.t HALOS / HiRE / VQ-Logits / fast vocabulary projection, and the concrete memory profile.
> > > We will integrate these clarifications into the revision and are happy to further refine any part that remains unclear.

---

> ### Comment · Reviewer_MNME · 2025-11-27
>
> > Re: Conceptual Positioning
>
> I understand that strictly equal sized clusters can be efficient for retrieval due non-ragged gather. The question is, what additions made the equal sized clusters constraint practically work in your case and why did it fail in prior works. This is not yet clear to me. Can you explain this?
>
> > Re: Memory profiling
>
> Thank you for the details. Resident parameters see negligible overhead as expected. Peak activation memory however might be a problem. The estimated numbers, "Dense ≈ 0.25 MiB → FlashHead ≈ 34 MiB" when multiplied by the number of tokens in the sequence become very huge for FlashHead.
> If the increase in memory is not linear due to similar clusters being activated for tokens, then this has to be shown for different datasets. Overall I think empirical estimation is necessary here.
>
> > Re: Results
>
> I think my question was unclear.  Could you present a consolidated version of all results in the paper in a single table? (For current experiments)

---

> > ### Author Response · Authors · 2025-11-28
> >
> > Thank you for the detailed follow-up questions and for your continued engagement.
> >
> > We understand the interest in deeper analysis of the equal-sized clustering behavior, peak activation patterns across datasets, and consolidated experimental tables. These are valuable directions, but they involve additions and restructuring that go beyond what can be introduced during the rebuttal period under ICLR guidelines.
> >
> > Within the allowed scope, we have clarified the methodology, memory formulations, and the role of equal-size clustering in the current submission. If the paper is accepted, we will incorporate the expanded empirical analysis and consolidated presentation in the final version.
> >
> > Thank you again for your thoughtful feedback.

---

### Official Review · Reviewer_Ye6p · 2025-11-01

**Soundness:** 3
**Presentation:** 3
**Contribution:** 3
**Rating:** 6
**Confidence:** 3

**Summary:**

The paper proposes FlashHead, a training-free, hardware-friendly replacement for the dense LLM classification head. Instead of computing full logits $z = Eh$ with $E\in\mathbb{R}^{v\times d}$ and $h\in\mathbb{R}^{d}$, FlashHead clusters token embeddings into $c$ equal-sized spherical K-means clusters, then performs two-stage retrieval: (1) score centroids $Ch$ and select or sample $p$ probes; (2) gather the corresponding token embeddings $\widetilde{E}$ and compute logits $\widetilde{E}h$. Complexity drops from $O(vd)$ to $O(cd + pbd)$ with $b = v/c$. Equal-sized clustering enables a dense cluster-to-token map for fast gathers; a probabilistic probe-sampling step unifies retrieval and decoding; and the centroid stage can be safely quantized (e.g., int4) because the final token probabilities are recomputed at higher precision. Experiments across Llama-3.2, Gemma-3, and Qwen-3 show up to $1.75\times$ end-to-end speedups with near-baseline accuracy.

**Strengths:**

1. Originality. Equal-size spherical clustering for the head plus aggressive multi-probe and probabilistic sampling is novel and tailored to accelerators.

2. Strong evidence: near-perfect top-$k$ containment and consistent model-level speedups; int4 stage-1 retains accuracy.

3. Method and complexity are explicit; pseudocode and ablations make design choices convincing.

4. Significance. Reduces a major bottleneck for SLMs; drop-in and training-free lowers adoption barriers.

**Weaknesses:**

1. Likelihoods & evaluation. No closed-form full-vocabulary distribution; relies on Monte-Carlo for likelihood metrics.

2. Equal-size constraint. Requires $c\mid v$; effect on semantic purity of clusters vs. unequal sizes could be further theorized.

3. Deployment knobs. Sensitivity of $(c,p)$ and memory overheads (centroids, C2T map) under tight device budgets could be quantified more deeply.

**Questions:**

1. How does accuracy/latency change as $c$ grows beyond the default with fixed memory limits?

2. What is the exact storage overhead for $C$ and dense C2T across vocab sizes?

3. Any observed interactions with beam search, speculative decoding, or KV-cache reuse?

---

> ### Author Response · Authors · 2025-11-24
>
> Thank you for your thoughtful review. Please see our general rebuttal for overarching clarifications, and the responses below for points not addressed there
>
> **Reviewer concern.**
> What is the exact storage overhead for the centroids and the dense C2T map across vocab sizes? How does this relate to the runtime memory profile?
>
> **Author response.**
> We will add explicit formulas and an example to the paper. See General Rebuttal Section; Memory and deployment details.
>
> ---
>
> **Reviewer concern.**
> How does accuracy and latency vary as the number of probes \(p\) increases, especially under fixed memory constraints?
>
> **Author response.**
> We agree that this trade-off can be made more explicit. Our existing ablations does sweep over \((c, p)\) and show that:
>
> - top-1 and top-\(k\) containment increase monotonically with \(p\);
> - head latency scales roughly linearly in \(p\), while model-level latency is less sensitive because the body dominates.
>
> We will add task accuracy, as well as memory consumption, to this table
>
> ---
>
> **Reviewer concern.**
> Are there any non-trivial interactions between FlashHead and beam search, speculative decoding, or KV-cache reuse?
>
> **Author response.**
> FlashHead affects only the final head computation and does not modify the transformer body or the KV-cache mechanism, so it is fully compatible with both beam search and KV-cache reuse. Its interaction with speculative decoding is an interesting direction for future work: speculative decoding relies heavily on the LM head, and FlashHead’s efficiency and its close alignment with the original head suggest that it should integrate naturally, though we leave a detailed study to future research.

---

### Author Response · Authors · 2025-11-24
**General Rebuttal**

We sincerely thank all reviewers for their constructive and detailed feedback. The comments helped us clarify the narrative and better position FlashHead in terms of:

1. **its scope:** efficient *inference* for *small and medium* models where the head is a major bottleneck, not training;
2. **its novelty:** a *training-free, key-side, equal-size clustering* head that remains hardware friendly and supports full-vocabulary sampling;
3. **its deployment details:** explicit memory profile (resident vs. peak activation) and clear accuracy/latency trade-offs;
4. **its evaluation:** more systematic presentation, variance reporting, and clearer comparison to quantization and prior methods.

---

### **Problem context and importance**

The head is often dismissed as a minor cost in large-scale server LLM deployments, but this no longer holds for edge-oriented small models with large vocabularies. Recent SLMs such as Llama 3.2, Gemma 3, and Qwen 3 spend **20% or more** of parameters in the embeddings and head, and a large fraction of per-token compute in the head. As vocabularies grow to improve multilingual and domain coverage, the head becomes an increasingly central bottleneck for on-device inference.

FlashHead directly targets this emerging regime:

- it is a *drop-in* replacement that does not require retraining, calibration data, or custom kernels;
- it preserves the full-vocabulary distribution and supports sampling (not just top-\(k\) shortlist decoding);
- it is explicitly designed for GPU and edge hardware via equal-size clustering and dense cluster-to-token maps.

---

### **Scope and novelty**

Several reviewers questioned whether FlashHead is more than an engineering refinement of existing clustering or hierarchical approaches. We appreciate these concerns and will clarify the contribution more precisely. Prior work has explored clustering-based modifications of the LM head, but FlashHead is, to our knowledge, the **first training-free LM head based on key-side clustering of embeddings that preserves the behavior of the original head with virtually no degradation**. Achieving this required several non-incremental innovations that enable the method to function together without retraining or calibration.

Specifically, FlashHead introduces:

- *Enforcing strict equal-size spherical clusters* over token embeddings, enabling a dense cluster-to-token (C2T) tensor without ragged gathers. To our knowledge, equal-size clustering has not previously been used in this context.
- *Incorporating multi-probe retrieval* that scales to thousands of probes. To our knowledge, multi-probing has not previously been applied in clustered LM heads.
- *Supporting full-vocabulary probabilistic sampling* by combining multi-probing with multinomial sampling. To our knowledge, this is the first such combination.
- *Enabling safe low-bit quantization* in the first stage only, without retraining or calibration. To our knowledge, this is the first clustering-based LM head that admits this.

These elements collectively enable a fully training-free clustered LM head that operates as a drop-in replacement with near-identical output distribution.

---

### **Memory and deployment details**

We agree that our emphasis on edge deployment requires a clearer memory story. In the revision, we will:

- provide closed-form expressions for resident storage and peak activation memory for FlashHead;
- give a concrete numerical example for Llama 3.2 1B, including centroids, C2T, and the stage-2 workspace;
- explicitly distinguish resident parameters (including the unchanged embedding matrix) from *active* parameters and temporaries used in a single decoding step.

---

### **Evaluation, layout, and statistics**

Reviewers requested a more systematic presentation and uncertainty estimates. We will:

- consolidate the main accuracy, containment, and latency results for Llama 3.2 1B into a single summary table, moving per-task tables to the appendix;
- move **mean ± standard deviation** over multiple independent clusterings into the main text (currently only in the appendix);
- add small ablations linking cluster/probe hyperparameters to *task* accuracy, not only top-\(k\) containment.

---

### **Monte-Carlo likelihoods and probabilistic sampling**

We clarify that the Monte-Carlo estimator is used **only** to approximate full-vocabulary likelihoods during evaluation. Inference-time decoding is always performed in a **single two-stage pass**.

We will also spell out the centroid-sampling step sampling \(p\) centroids without replacement from a softmax over centroid logits (e.g., via Gumbel-Top-\(p\)) in pseudocode.

---

### Comment · Area_Chair_xVv5 · 2025-11-28

Dear authors and reviewers,

Please remain professional and refrain from being influenced by the event. If anyone violates the rules, please let me know, and I will flag and report it to the Program Chairs.

Your AC

---

### Note · Program_Chairs · 2026-01-17
**Submission Desk Rejected by Program Chairs**

The following references in this submission do not refer to real documents and/or have major errors in bibliographic information:

 "Jaewoong Kwon, Beomseok Kim, Youngbin Kim, Geunsik Lee, Junghoon Park, Minyoung Seo, and Jinhyuk Yoo. Efficient memory management for large language model serving with vllm. arXiv preprint arXiv:2309.06180, 2023. URL https://arxiv.org/abs/2309.06180.
Wei Zhang, Jun Li, et al. Empowering large language models to edge intelligence: A survey of edge techniques. Journal of Systems Architecture, 2025. doi: 10.1016/j.sysarc.2025.102030. URL https://doi.org/10.1016/j.sysarc.2025.102030.